# Protocatechuic Acid and Syringin from *Saussurea neoserrata* Nakai Attenuate Prostaglandin Production in Human Keratinocytes Exposed to Airborne Particulate Matter



Myeongguk Jeong [1,†] , Yeongdon Ju [1,2,†] , Hyeokjin Kwon [1] , Yeeun Kim [1] , Kyung-Yae Hyun [3,*] and Go-Eun Choi [1,*]

[1] Department of Biomedical Laboratory Science, College of Health Sciences, Catholic University of Pusan, Busan 46252, Republic of Korea; audrnr04@gmail.com (M.J.); lrdrlr@naver.com (Y.J.); ghy8627@gmail.com (H.K.); yeeun0509@naver.com (Y.K.)

[2] Medical Science Research Center, Pusan National University, Yangsan 50612, Republic of Korea

[3] Department of Clinical Laboratory Science, Dong-Eui University, Busan 47340, Republic of Korea

[*] Correspondence: kyhyun@deu.ac.kr (K.-Y.H.); gechoi@cup.ac.kr (G.-E.C.); Tel.: +82-51-890-2683 (K.-Y.H.); +82-51-510-0563 (G.-E.C.)

[†] These authors contributed equally to this work.

**Abstract:** *Saussurea neoserrata* Nakai offers a reliable and efficient source of antioxidants that can help alleviate adverse skin reactions triggered by air pollutants. Air pollutants, such as particulate matter (PM), have the ability to infiltrate the skin and contribute to the higher occurrence of cardiovascular, cerebrovascular, and respiratory ailments. Individuals with compromised skin barriers are particularly susceptible to the impact of PM since it can be absorbed more readily through the skin. This study investigated the impact of protocatechuic acid and syringin, obtained from the n-BuOH extract of *S. neoserrata* Nakai, on the release of $PGE_2$ and $PGD_2$ induced by $PM_{10}$. Additionally, it examined the gene expression of the synthesis of $PGE_2$ and $PGD_2$ in human keratinocytes. The findings of this research highlight the potential of utilizing safe and efficient plant-derived antioxidants in dermatological and cosmetic applications to mitigate the negative skin reactions caused by exposure to air pollution.

**Keywords:** *Saussurea neoserrata* Nakai; preparative liquid chromatography; particulate matter; antioxidant; protocatechuic acid; syringin





## 1. Introduction

According to the World Health Organization (WHO), air pollution is the most significant environmental health risk factor, and, in 2019, it caused more than 4.2 million deaths (https://www.who.int accessed on 19 December 2022). The primary air pollutants that pose severe health risks are particulate matter (PM), ozone ($O_3$), and nitrogen dioxide ($NO_2$) [1]. Of these, PM, the primary contributor to air pollution, is comprised of suspended solid and liquid particles found in the atmosphere. [2]. $PM_{10}$ and $PM_{2.5}$, with diameters less than 10 and 2.5 microns, respectively, can penetrate deeply into the lungs and bloodstream, increasing the risk of respiratory, cardiovascular, and cerebrovascular diseases [3–5].

The skin acts as a protective layer against external pollutants, but harmful environmental pollutants can still affect it. Individuals with impaired skin barriers are more susceptible to PM due to increased absorption through the skin [6,7]. PM can even disrupt the skin's barrier function, further facilitating drug absorption [8]. The infiltration of PM into the skin can worsen skin conditions such as atopic dermatitis [9] as well as contribute to premature aging [10] and hyperpigmentation [11]. PM exposure in conjunction with UV rays can have a synergistic adverse effect on the skin, causing photoaging and even skin cancer [12,13].

Research has demonstrated that PM in the air can cause oxidative stress and inflammation by producing reactive oxygen species (ROS). Moreover, cytokines and matrix metalloproteinases can be triggered in various cell models, including human dermal fibroblasts, epidermal keratinocytes, and reconstructed epidermis [14–17]. Airborne PM contains toxic elements, including heavy metals and polycyclic hydrocarbons, which exert pro-oxidative and pro-inflammatory impacts on tissues. However, the composition of PM can differ based on various factors such as location, altitude, and season [18–20]. $PM_{10}$ induces the production of ROS via the aryl hydrocarbon receptor/NADPH oxidase-dependent pathway, and recent studies have suggested that dual oxidase 2 is also involved in the ROS production by keratinocytes exposed to PM [21–25]. Furthermore, PM enhances the synthesis of the eicosanoid mediator prostaglandin (PG) E2 and diminishes the expression of filaggrin in human keratinocytes, resulting in impaired skin barrier function [21]. However, studies have demonstrated that eupafolin, derived from the medicinal plant phyla Nodiflora, can effectively hinder the expression of cyclooxygenase (COX)-2 and the production of $PGE_2$ in HaCaT keratinocytes when exposed to PM. Similarly, resveratrol, a polyphenol present in grapes and red wine, has also been observed to reduce PM-induced COX-2 expression and $PGE_2$ production in fibroblast-like synoviocytes, which are human cells with similarities to fibroblasts [26–29]. Therefore, the use of safe and effective antioxidants in dermatological and cosmetic approaches may help alleviate the negative skin reactions caused by PM exposure [30–32].

For thousands of years, plants have been the source of traditional medicine systems that continue to offer new remedies to humankind [33,34]. The genus *Saussurea* in the Asteraceae family is a large and diverse group of plants that has been widely used in traditional medicine for many purposes. The genus *Saussurea* has been studied for its anti-inflammatory, antibacterial, and protective effects against hydrogen peroxide-induced cellular damage [35–37]. In Korea, the *Saussurea* genus is highly diverse with 32 recognized species, 16 of which are endemic [38]. Syringin is a glucoside present in a variety of plant species [39]. It has also been extensively studied for potential therapeutic effects because of its anti-inflammatory properties in human cell lines, anti-cancer properties in cell lines, and neuroprotective properties in rat models [40–42]. The aim of this study is to investigate whether the extracts of *S. neoserrata* Nakai, as well as protocatechuic acid and syringin, could affect the release of $PGE_2$ and $PGD_2$ induced by $PM_{10}$ and whether it could also affect the gene expression of the enzymes concerned with $PGE_2$ and $PGD_2$ synthesis in human keratinocytes.

## 2. Materials and Methods

### 2.1. Materials and Reagents

Preparative liquid chromatography (prep-LC) was performed using an LC-Forte/R system (YMC, Kyoto, Japan) equipped with a YMC-DispoPack cartridge (ODS, 30 g) (YMC, Kyoto, Japan). For column chromatography, Silica gel (230–400 mesh) (Merck, Darmstadt, Germany), Kromasil 100-5-C18 (Nouryon, Bohus, Sweden), and Sephadex LH-20 (Merck, Darmstadt, Germany) were used. Chromatography was performed on a Kiesel gel 60 F254 thin-layer chromatography (TLC) plate (Merck, Darmastdt, Germany) and RP-18 F254s TLC plates (Merck, Darmastdt, Germany); detection was performed via being sprayed with 10% $H_2SO_4$ solution and heated with a UV (254 nm and 356 nm). $^1H$, $^{13}C$ NMR, and 2D NMR experiments were performed on a Bruker AVANCE II 400 (400 MHz for $^1H$ NMR and 100 MHz for $^{13}C$ NMR) NMR spectrometer (Bruker, Karlsruhe, Germany) [43]. MS data was obtained by the Agilent 6530 accurate-mass Q-TOF LC–MS instrument (Agilent Technologies, Santa Clara, CA, USA) [44]. The information of isolated compounds from *S. neoserrata* Nakai was obtained from NCBI PubChem database (https://pubchem.ncbi.nlm.nih.gov/ accessed on 30 June 2023). All chemical reagents were purchased from commercial suppliers, Sigma Chemical Company (St. Louis, MO, USA).

## 2.2. Extraction and Isolation

*S. neoserrata* Nakai was purchased as 1 kg of dried whole plant from a local store (Daum International, Hanam, Republic of Korea). Dried whole plants were immersed in distilled water at a sample-to-solvent ratio of 1:20 (*w/v*) for 10 h at 90 °C. The extracts of *S. neoserrata* Nakai were filtered with filter paper and concentrated using a rotary evaporator. *S. neoserrata* Nakai samples were freeze-dried for 24 h and were stored at 4 °C. The freeze-dried extract was partitioned with EtOAc, n-BuOH, and water extract [45,46]. A flow chart of the entire extraction process is shown in Figure 1.

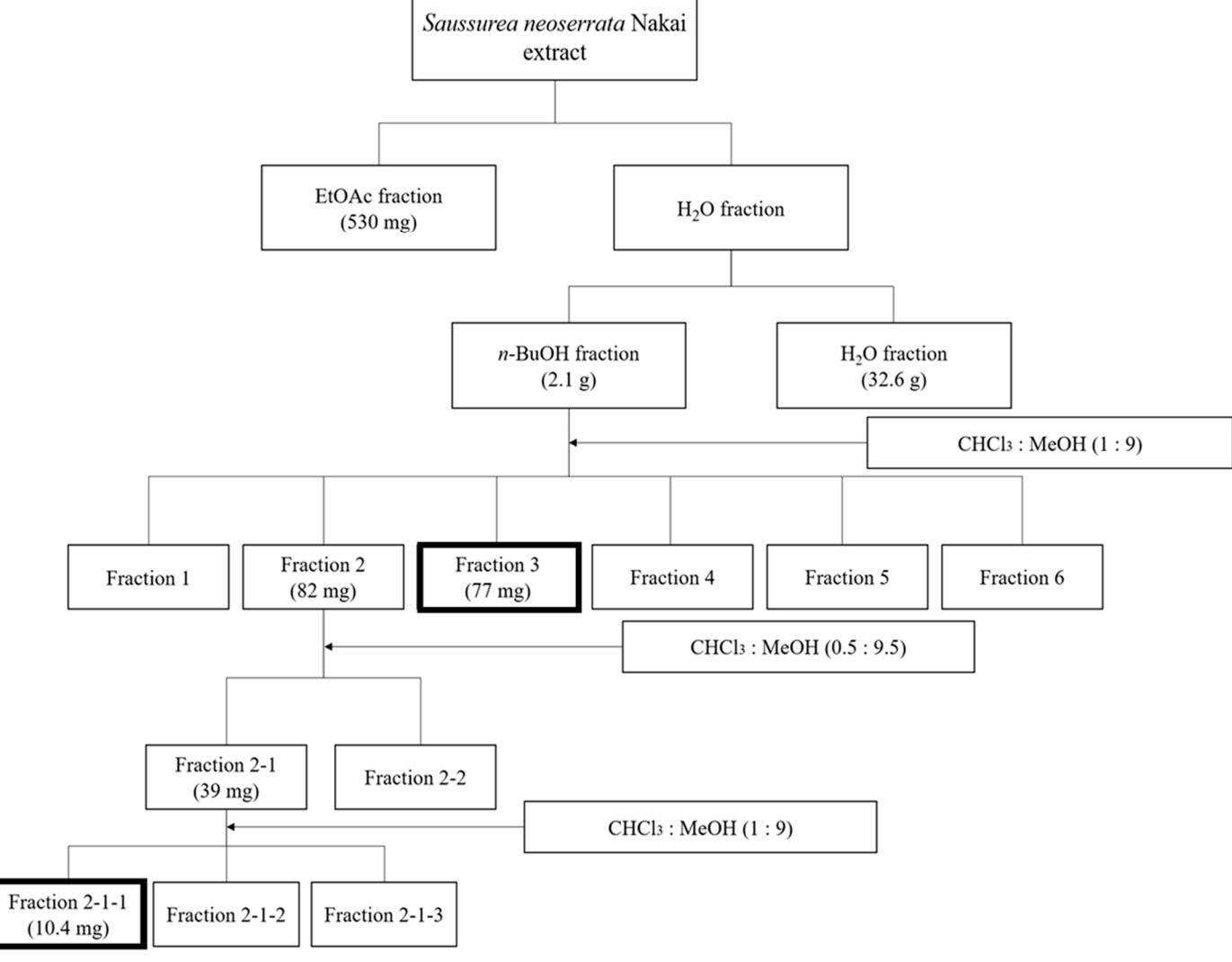

**Figure 1.** Schematic representation of the extraction and fractionation process of *S. neoserrata* Nakai. Bold squares represent the fraction for which each single compound was identified using NMR spectra.

## 2.3. Cell Culture

HaCaT cells are from an immortalized human keratinocyte cell line established by Norbert E. Fusenig and named so as to denote its origin from human adult skin keratinocytes that were propagated under low $Ca^{2+}$ conditions and elevated temperatures. The cells were cultured in a closed incubator at 37 °C in humidified air containing 5% $CO_2$. Cells were grown in DMEM medium (GIBCO, Paisley, UK) containing 10% fetal bovine serum, 100 U/mL penicillin, 100 μg/mL streptomycin, and 0.25 μg/mL amphotericin B [47].

## 2.4. Treatment of Cells with PM$_{10}$

The cells were plated on a 6-well culture plate at a density of $8 \times 10^4$ cells/well, and then cultured for 24 h. Under all experimental conditions, standardized PM$_{10}$-like fine dust (European Reference Material ERM-CZ120PM10) (Sigma Chemical Co., MO, USA) was mixed with the phosphate-buffered saline (PBS) at an appropriate concentration. The cells used in the experiments were exposed to PM$_{10}$ at various concentrations from 6.25 to 100 µg/mL for 24 to 48 h, respectively, depending on the measurement method. Additionally, the cells were treated with specified concentrations of *S. neoserrata* Nakai extract or dieckol, either in combination with PM$_{10}$ or separately. N-acetyl cysteine (NAC) (Sigma Chemical Co., MO, USA) was used as a positive control, and the antioxidant effect was evaluated [48].

## 2.5. Cell Viability Assay

Cell viability was assessed using the 3-[4,5-dimethylthiazol-2-yl]-2,5-diphenyl tetrazolium bromide (MTT) assay according to the reference [48]. Briefly, 200 µL of culture medium containing 1 mg/mL MTT (Amresco, OH, USA) was added to the cells and reacted for 2 h. After removing the medium, formazan crystals were dissolved in 200 µL of dimethyl sulfoxide (DMSO). Absorbance was measured at 595 nm using a microplate reader (BMG LABTECH GmbH, Ortenberg, Germany).

## 2.6. Enzyme-Linked Immunosorbent Assay (ELISA)

Production of PGE$_2$ by the cells was measured using the ELISA kit for detecting PGE$_2$ (Cayman Chemical Co., MI, USA). In the ELISA kit, a constant amount of PGE$_2$-acetylcholinesterase (AChE) conjugate is used as the PGE$_2$ tracer. Binding of AChE conjugates to monoclonal antibodies to PGE$_2$ uses the principle that the amount of PGE$_2$ present in a sample is inversely proportional. We measured this by referring to the manufacturer's instructions and reference [48].

Briefly, 50 µL of a 4-fold diluted cell culture or standard PGE$_2$ solution was added to microplate wells containing immobilized goat polyclonal anti-mouse IgG. PGE$_2$ tracer and PGE$_2$ monoclonal antibody were added and reacted at 4 °C for 18 h. After washing the wells, a solution of Ellman's reagent consisting of acetylthiocholine and 5,50-dithiobis-(2-nitrobenzoic acid) was added. After 60 min of reaction, absorbance was measured at 405 nm using a microplate reader (BMG LABTECH GmbH, Ortenberg, Germany). The concentration of PGE$_2$ was estimated using a calibration curve for a standard PGE$_2$ solution.

## 2.7. Assay for Cellular ROS Production

Cellular ROS levels were determined as previously described [49]. To measure the changes in intracellular active oxygen concentration, HaCaT cells were loaded with 5 µM chloromethyl derivative of 2′,7′-dichloro-dihydrofluorescein diacetate (CM2-DCFDA, Molecular Probes, Eugene, OR, USA). The change in ROS levels was measured using a flow cytometer (Becton Dickinson, OR, USA).

## 2.8. Analysis of Quantitative Reverse Transcription Polymerase Chain Reaction (qRT-PCR)

Analyses of the qRT-PCR were determined as previously described with some modifications [49]. In brief, total RNA was extracted from cells using the RNeasy kit (Qiagen, CA, USA), and this RNA was transferred to the High Capacity cDNA Archive Kit (Applied Biosystems, CA, USA). The qRT-PCR mixture (20 µL) consisted of SYBR® Green PCR Master Mix (Applied Bio-systems, CA, USA), cDNA (60 ng) and the specific primer sets (2 pmol). The specific primers were purchased from Macrogen (Seoul, Republic of Korea), and the sequences of primers are shown in Table 1. The mRNA levels were determined using the StepOnePlus Real-Time PCR System (Applied Biosystems, CA, USA) and were calculated relative to GAPDH (glyceralde-hyde 3-phosphate dehydrogenase).

**Table 1.** Sequences of primers used for the quantitative reverse transcription polymerase chain reaction (qRT-PCR) of the gene transcripts.

| Gene Name | GenBank Accession Number | Primer Sequences | Ref. |
|---|---|---|---|
| Cyclooxygenase 1 (COX-1)/Prostaglandin-endoperoxide synthase 1 (PTGS1) | NM_000962.4 | Forward: 5′-CAGAGCCAGATGGCTGTGGG-3′<br>Reverse: 5′-AAGCTGCTCATCGCCCCAGG-3′ | [50] |
| Cyclooxygenase 2 (COX-2)/Prostaglandin-endoperoxide synthase 2 (PTGS2) | NM_000963.3 | Forward: 5′-CTGCGCCTTTTCAAGGATGG-3′<br>Reverse: 5′-CCCCACAGCAAACCGTAGAT-3′ | [51] |
| Microsomal prostaglandin E synthase 1 (mPGES-1)/Prostaglandin E synthase (PTGES) | NM_004878.5 | Forward: 5′-AACCCTTTTGTCGCCTG-3′<br>Reverse: 5′-GTAGGCCACGGTGTGT-3′ | [52] |
| Microsomal prostaglandin E synthase 2 (mPGES-2); Prostaglandin E synthase 2 (PTGES2) | NM_025072.7 | Forward: 5′-GAAAGCTCGCAACAACTAAAT-3′<br>Reverse: 5′-CTTCATGGCTGGGTAGTAG-3′ | [52] |
| Cytosolic prostaglandin E synthase (cPGES)/ Prostaglandin E synthase 3 (PTGES3) | NM_006601.6 | Forward: 5′-ATAAAAGAACGGACAGATCAA-3′<br>Reverse: 5′-CACTAAGCCAATTAAGCTTTG-3′ | [52] |
| L-PGDS (PTGDS) | NM000954 | Forward: 5′-AACCAGTGTGAGACCCGAAC-3′<br>Reverse: 5′-AGGCGGTGAATTTCTCCTTT-3′ | [53] |
| H-PGDS (HPGDS) | NM014485 | Forward: 5′-CCCCATTTTGGAAGTTGATG-3′<br>Reverse: 5′-TGAGGCGCATTATACGTGAG-3 | [53] |
| GAPDH (glyceraldehyde 3-phosphate dehydrogenase) | NM_002046.3 | Forward: 5′-ATGGGGAAGGTGAAGGTCG-3′<br>Reverse: 5′-GGGGTCATTGATGGCAACAA-3′ | [54] |

*2.9. Statistical Analysis*

The data are presented as the mean ± standard deviation (SD) obtained from three or more independent experiments. Statistical analysis was performed using SigmaStat v.3.11 software (Systat Software Inc., CA, USA) with one-way analysis of variance (ANOVA). Subsequently, Dunnett's test was employed to compare all treatment groups against a single control group. A *p*-value below 0.05 was considered statistically significant.

**3. Results**

*3.1. Isolation of Protocatechuic Acid from S. neoserrata Nakai*

In the $^{1}$H-NMR spectrum, $\delta_H$ 7.46 (1H, br.s, H-2), $\delta_H$ 7.44 (1H, dd, *J* = 8.0, 2.0 Hz, H-6), $\delta_H$ 6.81 (1H, d, *J* = 8.0 Hz, In H-5), and three olefin methine proton signals were identified. Through this, the existence of a 1,3,4 trisubstituted benzene ring was predicted.

Seven carbon signals were identified in the $^{13}$C-NMR spectrum. Carbonyl carbon was confirmed in the signals of $\delta_C$ 170.8 (C-7), $\delta_C$ 151.3 (C-4), $\delta_C$ 145.9 (C-3), $\delta_C$ 124.0 (C-1), $\delta_C$ 123.9 (C-6), and $\delta_C$ 117.8 (C-5). Additionally, the presence of the 1,3,4-benzene ring was confirmed in the signal of $\delta_C$ 115.8 (C-2). Through this, the structure of the above compound 1 was determined as protocatechuic acid (Figure 2). ESI-MS was measured to confirm the molecular value. As a result, 153 [M-H]$^{-}$ was confirmed in negative mode, confirming the molecular weight of 154 g/mol. In the PubChem database, the molecular weight of proto-catechuic acid (PubChem ID: CID 72) was identified as 154.12 g/mol.

¹H-NMR spectrum

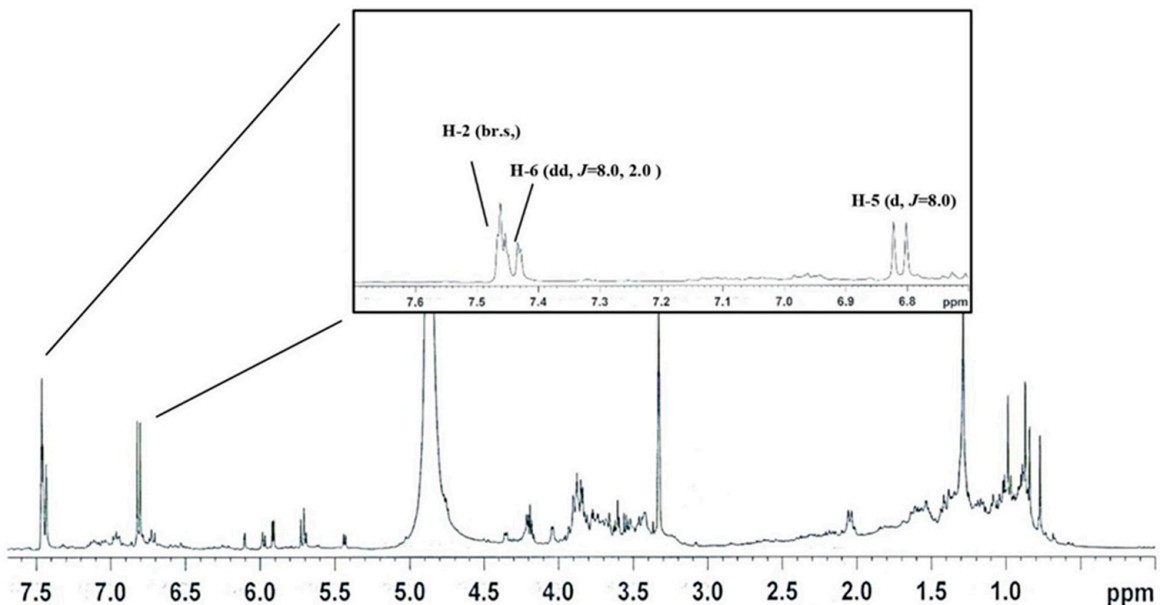

¹³C-NMR spectrum

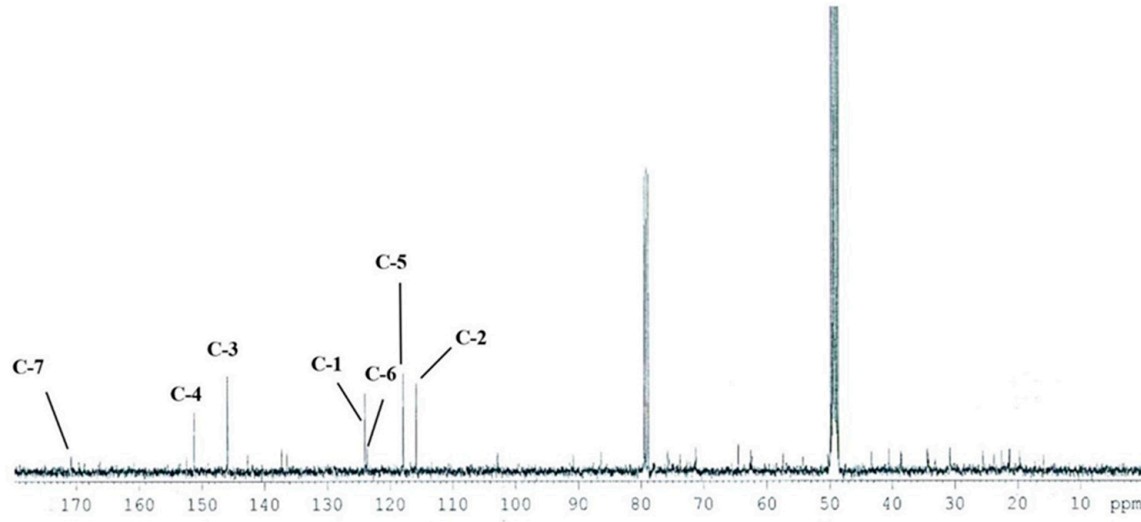

**Figure 2.** The ¹H-NMR and ¹³C-NMR spectrum of protocatechuic acid (77 mg).

### 3.2. Isolation of Syringin from S. neoserrata Nakai

In the ¹H-NMR spectrum, $\delta_H$ 6.87 (2H, br.s, H-2,6), $\delta_H$ 6.86 (1H, d, *J* = 15.6 Hz, H-7), and $\delta_H$ 6.59 (1H, dt, *J* = 15.6, 4.8 Hz, H-8), as well as the presence of one symmetric tetra-substituted aromatic ring and one trans-double bond, were confirmed. One molecule per molecule at $\delta_H$ 5.79 (1H, d, *J* = 6.4 Hz, H-1') and an oxygenated methylene signal at $\delta_H$ 4.57 (1H, d, *J* = 3.8 Hz, H-9) and $\delta_H$ 3.74 (6H, s, H-OCH₃) confirmed the methoxy signal. Based on this, the above compound was expected to be an aromatic glycoside derivative.

In the ¹³C-NMR spectrum, 17 carbon signals were identified, including one sugar molecule and two methoxy molecules. In the $\delta_C$ 154.3 (C-3,5) signal, oxygenated olefin quaternary carbon was confirmed and $\delta_C$ 136.4 (C-4).

One symmetric tetra-substituted aromatic ring and one trans-substituted ring in the presence of a double bond were confirmed. From the signal of $\delta_C$ 105.3 (C-1'), the anomer

carbon originating from the sugar's carbon 1 was confirmed. In addition, $\delta_C$ 79.2 (C-3′), $\delta_C$ 78.8 (C-5′), $\delta_C$ 76.5 (C2′), and $\delta_C$ were obtained from the sugar moiety signal of 72.0 (C-4′). Moreover, for $\delta_C$ 63.0 (C-6′), the sugar structure was determined as glucose.

One oxygenated methylene and two methoxy molecules were identified through the signals of $\delta_C$ 63.2 (C-9) and $\delta_C$ 57.0 (C-OCH₃). Finally, the structure of this compound 2 was determined as syringin (Figure 3). To confirm the molecular value, ESI-MS was measured. As a result, 417 [M-H + formic acid]⁻ was confirmed in negative mode, confirming the molecular weight of 372 g/mol. In the PubChem database, the molecular weight of syringin (PubChem ID: CID 5316860) was identified as 372.4 g/mol.

¹H-NMR spectrum

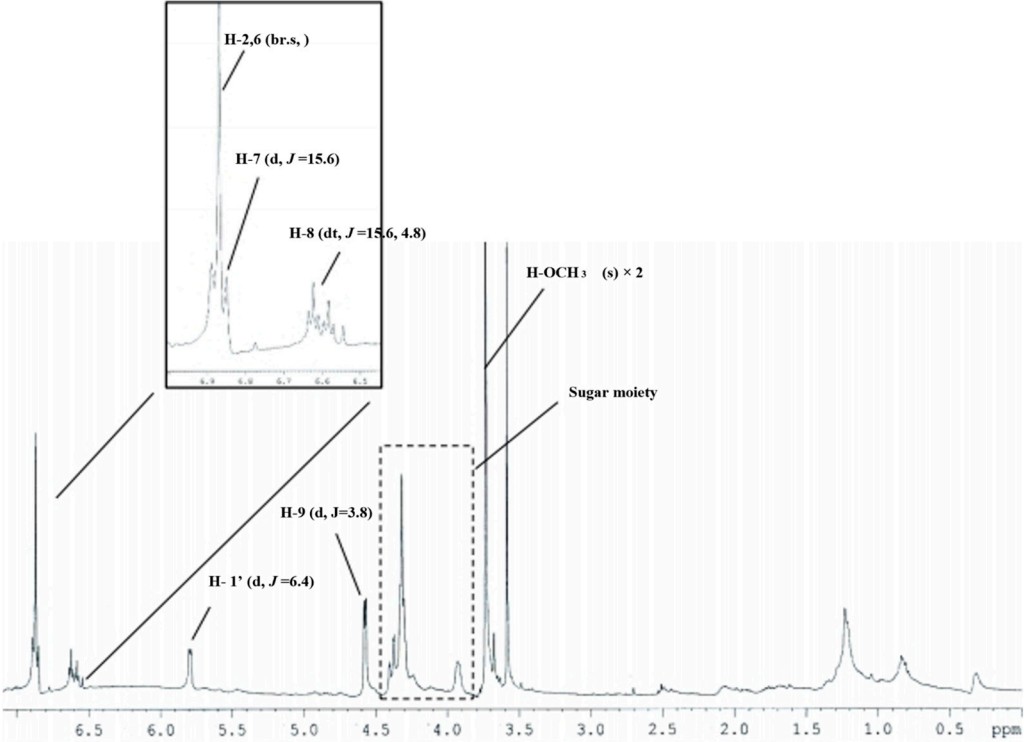

¹³C-NMR spectrum

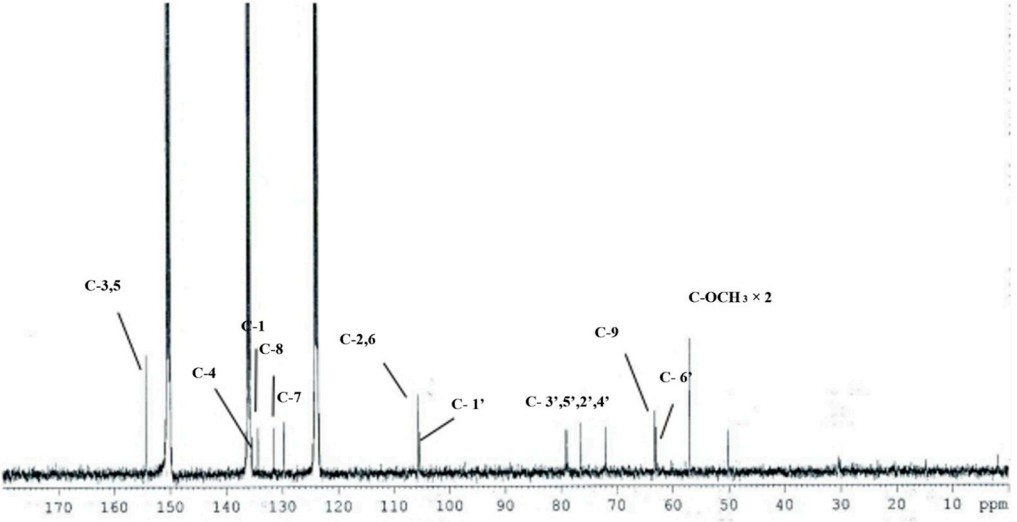

**Figure 3.** The ¹H-NMR and ¹³C-NMR spectrum of syringin (10.4 mg).

### 3.3. $PM_{10}$ Induces Cytotoxicity in $PGE_2$ and $PGD_2$ Release from Keratinocytes

To examine whether airborne $PM_{10}$ can cause cytotoxicity and inflammation, HaCaT cells were exposed to $PM_{10}$ in vitro. $PM_{10}$ treatments at 100 µg/mL for 48 h decreased the cell viability (Figure 4a). The conditioned cell culture media were used for determining the concentrations of $PGE_2$ and $PGD_2$. $PGE_2$ and $PGD_2$ production increased in the cells exposed to $PM_{10}$ at 25 µg/mL for 48 h (Figure 4b,c). To keep $PM_{10}$ within the non-toxic concentration range, a concentration of 12.5 µg/mL, which is greater than 95% cell viability, was used in the experiment.

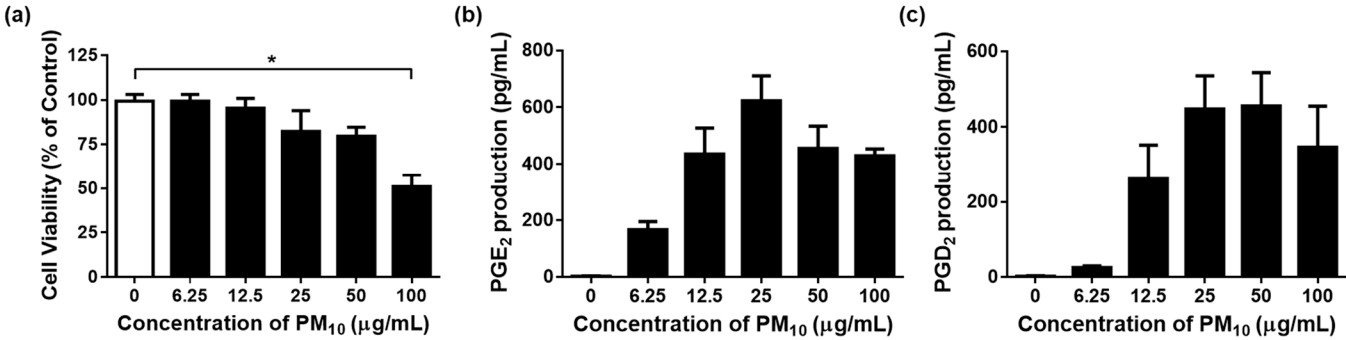

**Figure 4.** Effects of $PM_{10}$ on the viability and release of $PGE_2$ and $PGD_2$ in HaCaT keratinocytes were examined. The cells were exposed to different concentrations of $PM_{10}$ for a duration of 48 h in order to conduct the viability assay (**a**) and the $PGE_2$ (**b**) and $PGD_2$ (**c**) release assays. Control cells were treated with saline. Each bar represents the mean ± standard deviation (SD) (*n* = 4). All treatments were compared with the controls using one-way analysis of variance (ANOVA) followed by Dunnett's test * $p < 0.05$.

### 3.4. Effects of Various Concentrations of S. neoserrata Nakai Extracts on 12.5 µg/mL $PM_{10}$-Induced Cyto-Toxicity and $PGE_2$ and $PGD_2$ Release

To investigate whether the *S. neoserrata* Nakai extract had any effect on the viability of HaCaT keratinocytes, the cells were treated with various concentrations (10 to 200 µg/mL) of *S. neoserrata* Nakai extract for 48 h. Our results confirmed that the viability of HaCaT keratinocytes was not affected by *S. neoserrata* Nakai extract at the concentrations and culture durations tested (Figure 5a). Next, in order to confirm the effect of the *S. neoserrata* Nakai extract on the inflammatory response of HaCaT cells, the keratinocytes were exposed to 12.5 µg/mL $PM_{10}$ in the presence of 10 to 200 µg/mL *S. neoserrata* Nakai extract. The results indicate that the *S. neoserrata* Nakai extract inhibited $PM_{10}$-stimulated $PGE_2$ and $PGD_2$ release in a dose-dependent manner (Figure 5b,c).

### 3.5. Effects of Protocatechuic Acid and Syringin on PM10-Induced Keratinocyte Cytotoxicity and $PGE_2$ and $PGD_2$ Release

Protocatechuic acid showed a trend of decreasing HaCaT cell viability when added at concentrations between 30 and 100 µg/mL for 48 h (Figure 6a); however, the only statistically significant difference occurred at 100µg/mL of protocatechuic acid. Syringin did not alter the viability when tested at concentrations of up to 20 µg/mL, though there was a trend for decreasing HaCaT cell viability at concentrations above this (Figure 6b). In subsequent experiments, both protocatechuic acid and syringin were used at 10 and 20 µg/mL concentrations in order to keep within the non-toxic concentration range. Protocatechuic acid did not significantly reduce the release of $PGE_2$ (Figure 6c). However, syringin dose-dependently and significantly reduced the release of $PGE_2$ from keratinocytes exposed to 12.5 µg/mL $PM_{10}$ (Figure 6d). On the other hand, protocatechuic acid dose-dependently and significantly reduced the release of $PGD_2$ from keratinocytes exposed to 12.5 µg/mL $PM_{10}$ (Figure 6e). However, syringin did not significantly reduce the release of $PGD_2$ (Figure 6f). NAC (10 µg/mL) was used as a positive control antioxidant.

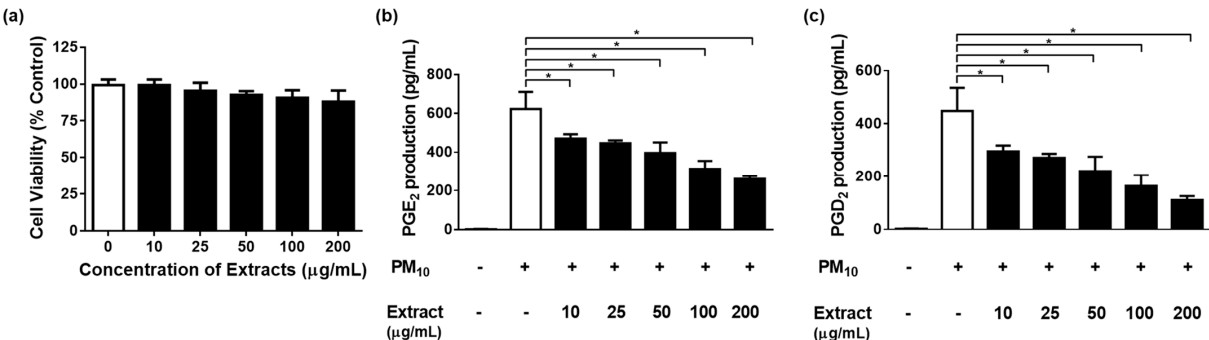

**Figure 5.** Effects of *S. neoserrata* Nakai extracts on the viability and the $PGE_2$ and $PGD_2$ release of HaCaT keratinocytes in response to $PM_{10}$. The cells were treated with 12.5 μg/mL $PM_{10}$ in the presence of various concentrations of *S. neoserrata* Nakai extract for 48 h for the purposes of a viability assay (**a**) and $PGE_2$ (**b**) and $PGD_2$ (**c**) release assays. Each bar represents the mean ± SD ($n = 4$). All treatments were compared with the $PM_{10}$–only control using one-way ANOVA followed by Dunnett's test * $p < 0.05$.

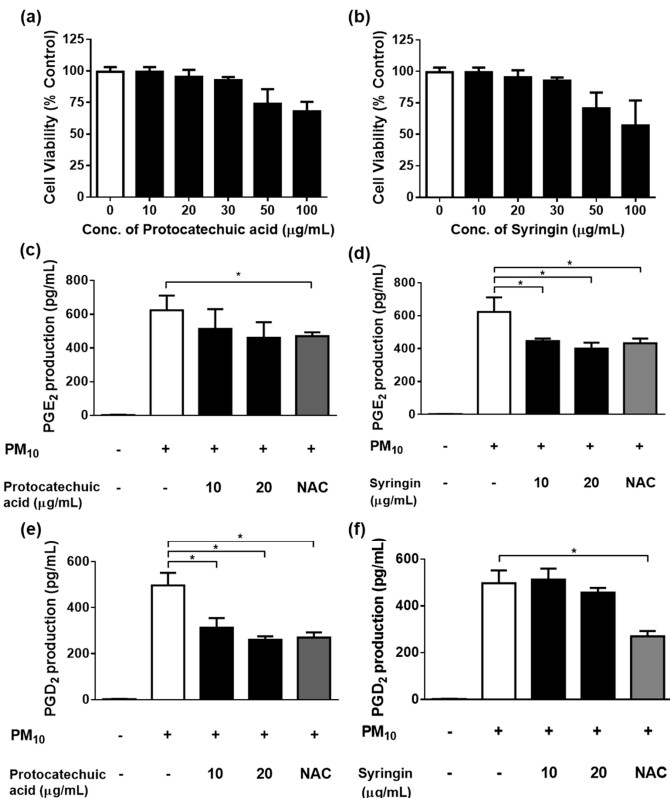

**Figure 6.** Effects of protocatechuic acid and syringin on $PM_{10}$–induced keratinocyte cytotoxicity and $PGE_2$ and $PGD_2$ release. HaCaT keratinocytes were treated with various concentrations of protocatechuic acid and syringin for 48 h, and the resulting cell viability was measured (**a**,**b**). (**c**) Cells were treated with 12.5 μg/mL $PM_{10}$ in the presence or absence of protocatechuic acid at the indicated concentrations for 48 h for the $PGE_2$ release assay. (**d**) Cells were treated with 12.5 μg/mL $PM_{10}$ in the presence or absence of syringin at the indicated concentrations for 48 h for the $PGE_2$ release assay. (**e**) Cells were treated with 12.5 μg/mL $PM_{10}$ in the presence or absence of protocatechuic acid at the indicated concentrations for 48 h for the $PGD_2$ release assay. (**f**) Cells were treated with 12.5 μg/mL $PM_{10}$ in the presence or absence of syringin at the indicated concentrations for 48 h for the $PGD_2$ release assay. NAC (10 μg/mL) was used as a positive control antioxidant in each assay. Each bar represents the mean ± SD ($n = 4$). All treatments were compared with the $PM_{10}$–only control using one-way ANOVA followed by Dunnett's test * $p < 0.05$.

### 3.6. Effects of Protocatechuic Acid and Syringin on PM₁₀-Induced ROS Production

Keratinocytes were treated with 12.5 µg/mL $PM_{10}$ to induce oxidative stress, and the ability of protocatechuic acid and syringin to remove ROS was measured. The ROS removal ability of the *S. neoserrata* Nakai extract was assessed using DCF-DA, the green fluorescence being proportional to the amount of ROS present. When the HaCaT cells were treated with 12.5 µg/mL $PM_{10}$, the intracellular ROS levels were increased compared with the control group. Cells with $PM_{10}$-induced increased ROS were treated with *S. neoserrata* Nakai extract (100 and 200 µg/mL), protocatechuic acid (10 and 20 µg/mL), and syringin (10 and 20 µg/mL). Each treatment decreased $PM_{10}$-induced ROS in a concentration-dependent manner. This experiment confirmed that treatment with the protocatechuic acid and syringin derived from *S. neoserrata* Nakai extracts effectively reduced the intracellular ROS levels in HaCaT keratinocytes (Figure 7).

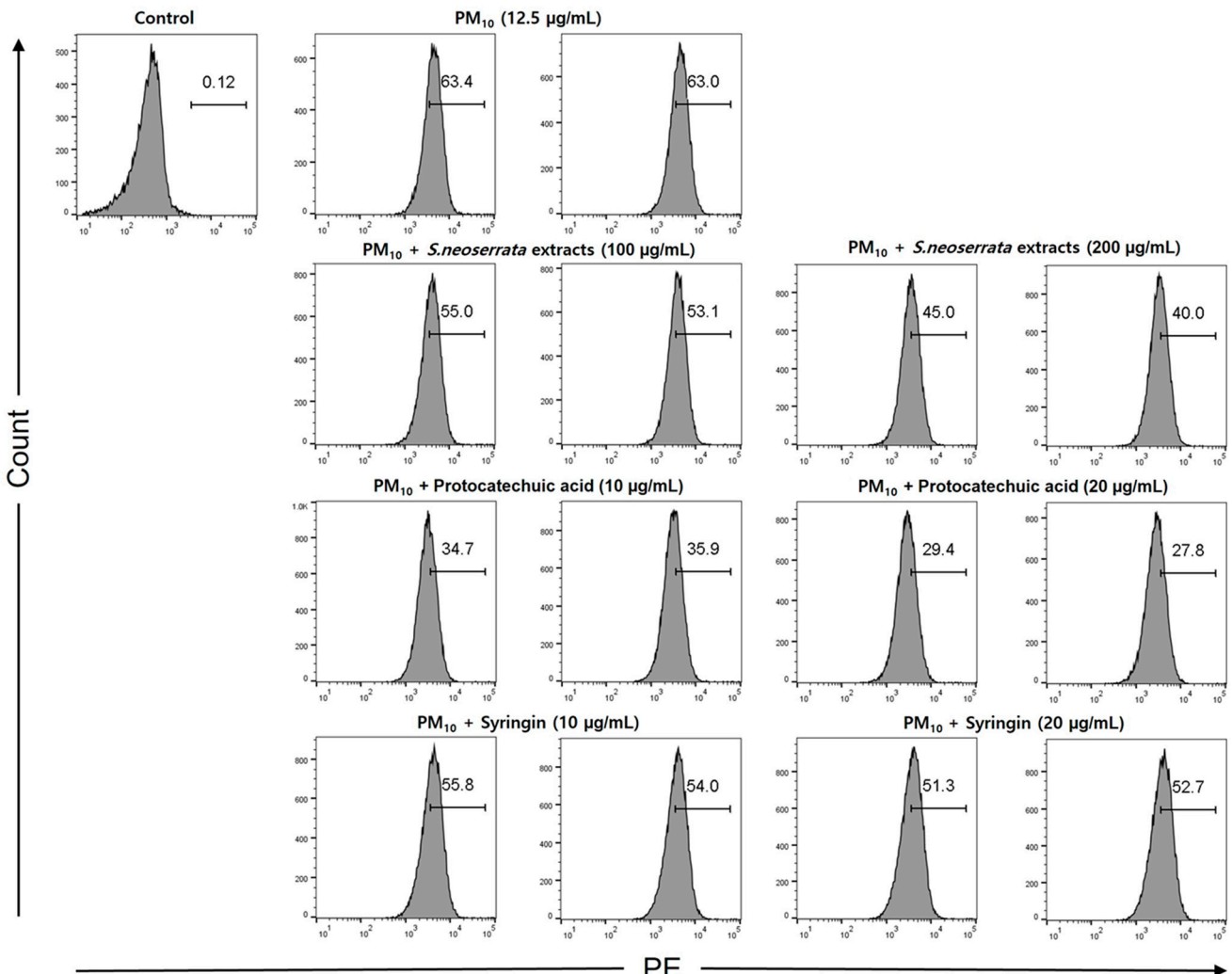

**Figure 7.** Effect of *S. neoserrata* Nakai extract, protocatechuic acid, and syringin on the $PM_{10}$-induced ROS production. Keratinocytes were exposed to a concentration of 12.5 µg/mL $PM_{10}$ for 24 h. The ROS production of keratinocytes was measured by flow cytometry using CM2-DCFDA. Compared to the control group, treatment with protocatechuic acid and syringin showed a concentration-dependent decrease in intracellular ROS production.

### 3.7. Effects of Protocatechuic Acid and Syringin on the $PM_{10}$-Induced Gene Expression of the Enzymes Involved in the $PGE_2$ and $PGD_2$ Synthesis

Since the $PM_{10}$-induced release of $PGE_2$ and $PGD_2$ was attenuated by *S. neoserrata* Nakai extract, additional experiments were performed to determine the mRNA expression levels of the enzymes involved in $PGE_2$ and $PGD_2$ production. NAC (10 µg/mL) was also tested in the same manner as a positive control antioxidant.

As the protocatechuic acid extracted from *S. neoserrata* Nakai extracts was more effective at inhibiting $PGD_2$ production, we aimed to investigate the mRNA expression levels of L-PGDS and H-PGDS, which are both involved in $PGD_2$ production, in response to protocatechuic acid. Treatment with 12.5 µg/mL $PM_{10}$ increased the mRNA expression levels of L-PGDS, and these changes were greatly attenuated by protocatechuic acid treatment. Treatment with 10 µg/mL NAC resulted in a similar level of inhibition as that observed using 20 µg/mL protocatechuic acid (Figure 8a). On the other hand, the H-PGDS expression increase induced by $PM_{10}$ treatment was not attenuated by protocatechuic acid. However, 10 µg/mL NAC did reverse the increase in H-PGDS expression (Figure 8b).

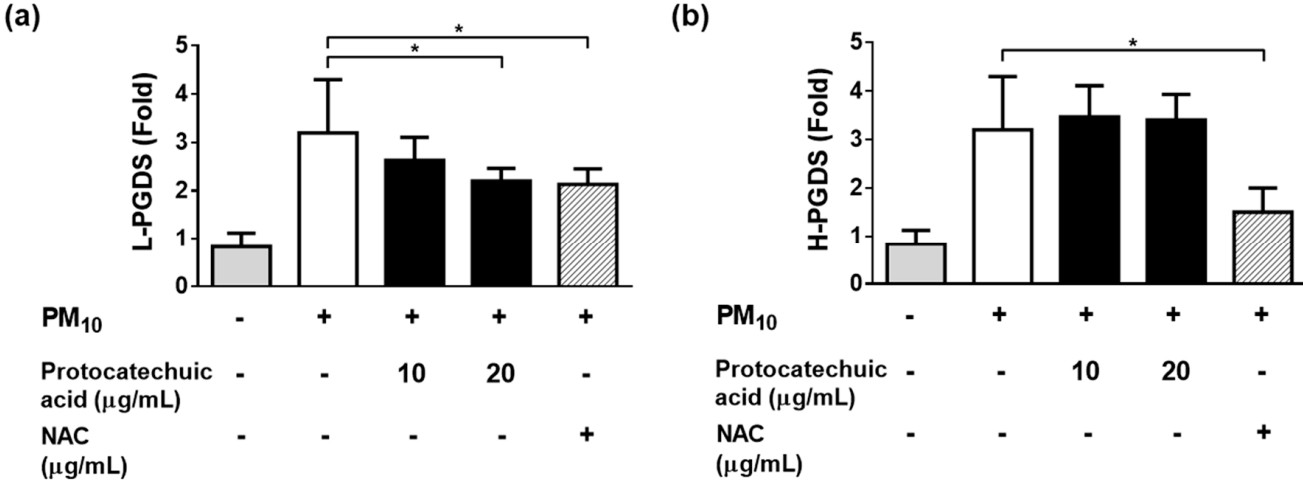

**Figure 8.** Effects of protocatechuic acid on the $PM_{10}$–induced gene expressions of the enzymes involved in $PGD_2$ synthesis. Cells were treated with 12.5 µg/mL $PM_{10}$ for 24 h in the presence or absence of protocatechuic acid at the indicated concentrations in order to determine the mRNA expression of enzymes involved in $PGD_2$ synthesis (L–PGDS and H–PGDS). Treatment with 20 µg/mL protocatechuic acid significantly reduced the expression of L–PGDS (**a**). However, H–PGDS expression was not reduced (**b**). N–acetyl cysteine (NAC) was employed as a positive control antioxidant. Each bar represents the mean $\pm$ SD ($n = 4$). All treatments were compared with the $PM_{10}$–only control using one-way ANOVA followed by Dunnett's test * $p < 0.05$.

The syringin extracted from *S. neoserrata* Nakai extracts was more effective in inhibiting $PGE_2$ production than other prostaglandins. Therefore, the correlation between the mRNA expression levels of the mPGES–1, mPGES–2, and cPGES (all involved in $PGE_2$ production) and treatment with syringin was investigated. Treatment with 12.5 µg/mL $PM_{10}$ increased the expression of mPGES-1 at the mRNA level. However, these changes were greatly attenuated by treatment with 20 µg/mL syringin (Figure 9a). A 10 µg/mL concentration of NAC suppressed the increased mPGES–1 expression to a similar level as 20 µg/mL syringin. $PM_{10}$ treatment at 12.5 µg/mL did not increase the expression of mPGES–2 and cPGES at the mRNA level (Figure 9b,c).

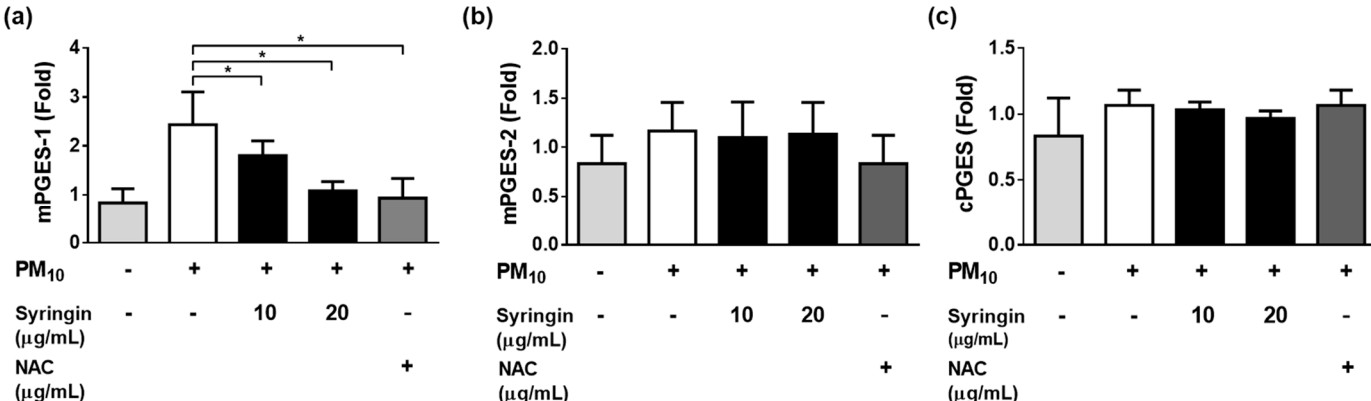

**Figure 9.** Effects of syringin on the $PM_{10}$–induced gene expressions of the enzymes involved in the $PGE_2$ synthesis. To examine the mRNA expression of enzymes associated with $PGD_2$ synthesis (L–PGDS and H–PGDS), cells were exposed to 12.5 µg/mL $PM_{10}$ for 24 h with or without syringin at the specified concentrations. Treatment with 12.5 µg/mL $PM_{10}$ increased the expression of mPGES-1 at the mRNA level. However, treatment with syringin decreased the expression of mPGES–1 in a concentration–dependent manner (**a**). PM10 treatment at 12.5 µg/mL did not significantly increase the expression of mPGES–2 and cPGES at the mRNA level (**b,c**). N–acetyl cysteine (NAC) was employed as a positive control antioxidant. Each bar represents the mean ± SD (*n* = 4). All treatments were compared with the $PM_{10}$–only control using one-way ANOVA followed by Dunnett′s test * *p* < 0.05.

## 4. Discussion

In the present paper, extracts from the plant species *S. neoserrata* Nakai, which have not been studied enough before, were examined, and compounds from these extracts were isolated. This paper is the first in the world to investigate the composition of *Saussurea neoserrata* Nakai using LC–MS/MS and NMR. Isolation results identified protocatechuic acid ($C_7H_6O_4$) and syringin ($C_{17}H_{24}O_9$). This is the first report on a structural explanation and identification of the protocatechuic acid and syringin active compounds contained in *S. neoserrata* Nakai extracts. The established preparative-liquid chromatography (prep-LC) method proved to be simple, precise, and accurate. The protocatechuic acid identified in 3,4-dihydroxybenzoic acid has been previously found to play an important role in antioxidant activity [55]. The mass spectrum of the 3,4-dihydroxybenzoic acid identified the molecular ion $[M-H]^-$ at a m/z 153 atomic mass unit (amu) and a base peak $[M-H-CO_2]^-$ at m/z 108.9 amu.

Yang CY et al. demonstrated that 5 µg/mL of *Artocarpus altilis* extract is an anti-inflammatory component to prevent PM-induced inflammation in HaCaT cells [56]. *Ecklonia cava* extract has been shown to mitigate $PM_{10}$-induced $PGE_2$ production [48]. In the present study, the *S. neoserrata* Nakai extracts served as the protective component against $PM_{10}$ toxicity for HaCaT keratinocytes. The *S. neoserrata* Nakai extracts no more effectively attenuated $PGE_2$ and $PGD_2$ production in the cells exposed to varying concentrations of $PM_{10}$ than NAC, which was used as a positive control antioxidant. The protocatechuic acid and syringin purified from *S. neoserrata* Nakai extracts also exhibited inhibitory activity against $PM_{10}$-induced $PGE_2$ and $PGD_2$ production.

Protocatechuic acid is present in a significant number of plants used in folk remedies, and results obtained in this study are supporting this usage [57]. The anti-inflammatory and antioxidant activity of protocatechuic acid are also proven [58]. Lipocalin-type prostaglandin D synthase (L-PGDS) is from a group of secreted proteins that make up the lipocalin superfamily and which bind to lipophilic molecules [59]. L-PGDS is a monomeric protein present in several mammalian central nervous system tissues and the male genital organs; in contrary, it is abundant in cerebrospinal fluid [60]. Hematopoietic prostaglandin D synthase (H-PGDS) is a member of the sigma-class of glutathione-S-transferases (GST) [61]. H-PGDS is expressed in various immune and inflammatory cells, such as mast cells in rat

model and type 2 T lymphocytes in human cell lines [62,63]. L-PGDS and H-PGDS are involved in the production of prostaglandin $D_2$ ($PGD_2$), which is obtained from PGH2 via isomerization [64]. The relationship between $PGD_2$ and inflammation is complex and not fully established. However, $PGD_2$ has been reported to contribute to allergic inflammatory reactions [65,66].

The results of our study show that treatment with protocatechuic acid significantly reduces $PM_{10}$-induced increases in L-PGDS in a concentration-dependent manner. A decrease in L-PGDS, one of the synthetases involved in $PGD_2$ production, eventually leads to a reduction in PGD2 levels. This finding suggests that protocatechuic acid may have a protective role in inflammation and oxidative stress associated with $PM_{10}$ exposure.

The synthesis of $PGE_2$ initiates by transforming membrane phospholipids into arachidonic acid with the assistance of phospholipase A2. Subsequently, arachidonic acid undergoes chemical reactions to form $PGG_2$, which further converts to $PGH_2$. These conversion processes are facilitated by the enzymes COX-1 and COX-2 [67]. Both COX isoforms can be detected in healthy human tissues and show increased expression in different disease states [68]. The transformation of PGH2 into $PGE_2$ is facilitated by enzymes such as mPGES-1, mPGES-2, and cPGES [69], with mPGES-1 being the primary isoform responsible for heightened $PGE_2$ synthesis during inflammatory processes [70].

Our results demonstrate that elevated mPGES-1 in response to $PM_{10}$ exposure was significantly suppressed by syringin treatment. This means that the synthesis of $PGE_2$ is consequently inhibited. These results may mean that syringin has potential as a treatment for the adverse health effects of $PM_{10}$ exposure. Further studies are required to investigate the mechanisms by which syringin shows $PGE_2$ inhibitory effects and to determine the safety and efficacy required for its clinical use.

Previous studies have demonstrated that various antioxidants, including NAC, can reduce the cellular ROS production induced by PM [71,72]. The ROS generated by PM can stimulate the MAPK family, which includes ERK, JNK, and p38 kinase, along with the NF-κB signaling pathway. This activation subsequently triggers redox-sensitive transcription factors like AP-1 and NF-κB [14,73,74]. COX-2 mRNA expression is controlled by different transcription factors, including the cyclic-AMP response element binding protein and NF-κB, which become activated by various MAPKs and other protein kinases [75]. When keratinocytes are exposed to PM, it can stimulate MAPKs such as ERK, p38, and JNK, resulting in the eventual expression of COX-2 [76].

## 5. Conclusions

The response to PM involves multiple redox-sensitive pathways that are involved in the regulation of $PGE_2$ and $PGD_2$ synthesis. It is suggested that the antioxidants present in *S. neoserrata* Nakai extracts, namely protocatechuic acid and syringin, may reduce the production of $PGE_2$ and $PGD_2$ by inhibiting signaling pathways in response to PM exposure in a concentration-dependent manner. However, further research is necessary to confirm this concept and to evaluate the effectiveness of protocatechuic acid and syringin in vivo.

**Author Contributions:** Conceptualization, M.J., Y.J., H.K., Y.K., K.-Y.H. and G.-E.C.; methodology, M.J., Y.J., H.K., Y.K., K.-Y.H. and G.-E.C.; validation, M.J., Y.J., H.K., Y.K., K.-Y.H. and G.-E.C.; investigation, M.J., Y.J., H.K., Y.K., K.-Y.H. and G.-E.C.; writing—original draft preparation, M.J., Y.J., H.K. and Y.K.; writing—review and editing, M.J., Y.J., K.-Y.H. and G.-E.C.; supervision, K.-Y.H. and G.-E.C.; project administration, K.-Y.H. and G.-E.C.; funding acquisition, K.-Y.H. and G.-E.C. All authors have read and agreed to the published version of the manuscript.

**Funding:** This work was supported by the academic research funds from the Catholic University of Pusan in 2021 and the National Research Foundation of Korea (NRF) grant funded by the Korean government (MSIT) (Nos. NRF-2022R1F1A1074419 and NRF-2022R1F1A1066041).

**Institutional Review Board Statement:** Not applicable.

**Informed Consent Statement:** Not applicable.

**Data Availability Statement:** All data generated and analyzed during this study are included in the main article and all related UTR links were provided within the article under relevant mentions.

**Conflicts of Interest:** The authors declare no conflict of interest.

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
