# Peer review of "Protocatechuic Acid and Syringin from Saussurea neoserrata Nakai Attenuate Prostaglandin Production in Human Keratinocytes Exposed to Airborne Particulate Matter"

_cimb, doi:10.3390/cimb45070376_

Round 1

Reviewer 1 Report

The manuscript is interesting, but some corrections and information need to be added, as mentioned below. Also, please review the included pdf file to review all suggested corrections.

1.      1. Paragraph 63-67 express it as objective

2.      Line 71. Add following: Place where the material was grown or harvested, state of the plant (green, dry), % Moisture, part of the plant that was used for the test (whole, leaves, stems, flowers, etc.)

3.      Paragraph 72-76 should be at the beginning of section 2.2

4.      Lines 87-88. Include as reference the mass spectra ecomponent libraries used for verify the purified protocatechuic acid

5.      Include references to: Paragraph 92-105, Paragraph 120-129, Paragraph 132-137, Paragraph 140-152, Paragraph 155-164

6.      Lines 201-202 and lines 228-230. Please, include Retention index and and the registry number assigned in the mass spectra library used.

7.      Improve the definition of images and  increase font size of Figures 7 Figure 8, Figure 9 and Figure 10. the units of concentration of PM10 are blurred. if it is the case, in the bars of the graphs of figures 6, 7 and 8 add letters that indicate if the treatments are the same or different according to the result of the Dunnett`s test.

8.      Figure 8, in c,d, e, f incises, these four tests were all without adding PM10?. In the description of this figure, line 290 indicate separately the descriptions of figures c, d, e, f

9.      Paragraph 295-305, Mention Figure 10, since a description of it is made

10.  Paragrpah 312-334. Why were only these primers tested (L-PGDS, H-PGDS, mPGES-1, cPGES ? Table 1 shows 8 primers. In this same paragraph, Why different amounts of PM10 are used? (12.5 μg/mL and 50 μg/mL ). Isn't it better to use a single concentration of PM10 to compare both effects? Include the explanation in the text

11.  Paragraph 357 429. In the discussion, it remains to compare the most important results with other similar studies, if any.

12.  Line 398. “….(US, MAHADEVA RAO)…” Is  a reference?, add its corresponding identifier number. If it is not, indicate what it means or remove it

Author Response

첨부 파일을 참조하십시오.

Reviewer 2 Report

Please follow instructions in reviewed word file given. 

Reviewer 3 Report

The Manuscript by M. Jeong, Y. Ju, H. Kwon, Y. Kim, K. Hyun, G.-E. Choi “Protocatechuic acid and syringin from Saussurea neoserrata Nakai attenuate prostaglandin production in keratinocytes exposed to airborne particulate matter” describes isolation and physicochemical characterization of several components of S. neoserrata Nakai extract as well as their effect on particulate matter-induced prostaglandins biosynthesis and release in human keratinocytes. Overall, biochemical part of the work is performed on a high level and results obtained are beyond any doubts. Though it remains unclear, what component of S. neoserrata Nakai extract causes the decrease in PGE2 and PGD2 production by cells exposed by PM10. Extraction and chromatographic separation of extractive compounds is described in detail. However, the assignment of the structure of syringin does not seem to be reliable. A large number of the signals of impurities appear in NMR spectra of syringin (as well as in the spectra of protocatechuic acid) presented in the manuscript. Moreover, the presence of ignal m/z 417 [M H + HCO2H]+ in mass-spectrum does not confirm molecular weight of 372. Authors should register NMR spectra of reference compounds and compare them with spectra obtained for protocatechuic acid and "syringin" presented in the manuscript or at least compare the spectra with published data. Comparison of HPLC chromatograms with literature data can be done as well. Compound described as "syringin" in the manuscript may turn out to have an other structure...

Several remarks have to be taken into account as well:

1.       The mobile phase composition presented in the manuscript (CHCl3–MeOH 1:9 and 0.5:9.5) probably have to be checked one more time. Maybe some kind of mistyping took place.

2.       In Section 2.6, there is incorrect term "The levels of PGE2 protein in the culture medium...". Prostaglandin is not a protein, so the phrase should be clarified.

3.       There are some missings in the references list (e.g., refs. 36, 37, 39).

I believe that the Paper by by M. Jeong, Y. Ju, H. Kwon and co-authors is interesting for the auditory of the Current Issues in Molecular Biology journal and can be published after appropriate revising of the title compounds structure.

Round 2

Reviewer 1 Report

Authors:

A better presentation and structuring of the manuscript is appreciated. Also, the review shows that most of the suggestions indicated in the first review were implemented. Even so, the following corrections are suggested:

 -Line 68. Replace “saussurea” with “Saussurea”

-Line 144. Replace “standard curve” with “calibration curve”. Incorporate data from the calibration curve, such as concentrations used to construct the curve and r2 for the straight line

-Line 158, Remove middle hyphen in "primer" word

-Homogenize the font size of the axes of all figures

Reviewer 3 Report

The structure of syringin is still not properly proved!

NMR spectra have to be compared with spectra of authentic samples or at least published data on chemical shifts. The presence of formic acid molecule in detected ion in mass-spectrometric analysis should be explained.

Otherwise, everything is well.
